# Intracranial Aneurysms and Lipid Metabolism Disorders: From Molecular Mechanisms to Clinical Implications

**DOI:** 10.3390/biom13111652

**Published:** 2023-11-14

**Authors:** Tonglin Pan, Yuan Shi, Guo Yu, Abdureshid Mamtimin, Wei Zhu

**Affiliations:** 1Department of Neurosurgery, Huashan Hospital, Shanghai Medical College, Fudan University, Shanghai 200090, China; 22211220055@m.fudan.edu.cn (T.P.); yuanshi18@fudan.edu.cn (Y.S.); yg18930819897@163.com (G.Y.); 22211220046@m.fudan.edu.cn (A.M.); 2Neurosurgical Institute, Fudan University, Shanghai 200032, China

**Keywords:** intracranial aneurysm, lipid metabolism disorder, statin, subarachnoid hemorrhage

## Abstract

Many vascular diseases are linked to lipid metabolism disorders, which cause lipid accumulation and peroxidation in the vascular wall. These processes lead to degenerative changes in the vessel, such as phenotypic transformation of smooth muscle cells and dysfunction and apoptosis of endothelial cells. In intracranial aneurysms, the coexistence of lipid plaques is often observed, indicating localized lipid metabolism disorders. These disorders may impair the function of the vascular wall or result from it. We summarize the literature on the relationship between lipid metabolism disorders and intracranial aneurysms below.

## 1. Introduction

Intracranial aneurysm (IA) is a localized dilatation or protrusion of the intracranial arterial wall, which is a common intracranial vascular lesion with a prevalence of approximately 3–6% [1]. IA has various clinical manifestations, ranging from asymptomatic to causing neurological dysfunction or compression of adjacent structures [2]. However, the most severe consequence is IA rupture leading to subarachnoid hemorrhage (SAH), which is a life-threatening acute cerebrovascular event with a mortality rate of up to 50% and a complication rate of up to 70% [3]. Therefore, preventing and treating IA formation and rupture is an important topic in intracranial vascular disease.

The pathological changes in intracranial aneurysms mainly include loss of the internal elastic layer, degradation of the extracellular matrix, and abnormal function of vascular smooth muscle cells. Currently, genetic factors, hemodynamic abnormalities, and various vascular risk factors are believed to participate in the formation process of IA. In the context of genetic susceptibility, the blood vessels at bifurcation sites are more prone to mechanical damage due to congenital structural weaknesses or local metabolic abnormalities under the impact of blood flow. The endothelial layer is the first to be compromised, followed by a series of changes within the vascular wall due to lipid deposition and the impact of blood flow. Inflammatory cells infiltrate and secrete inflammatory mediators, while smooth muscle cells undergo an inflammatory response, leading to phenotypic changes from a contractile to a secretory phenotype. With the alterations in the microstructure of the vessel wall, the influence of lipid deposition and the impact of blood flow further expand, intensifying the local inflammatory response. This results in apoptosis of endothelial cells and smooth muscle cells, ultimately weakening the mechanical strength within the vessel wall and causing localized outward bulging, forming an arterial aneurysm. As hemodynamics, inflammatory responses, and the local metabolic environment continue to change, the blood vessels gradually become weaker, promoting the development of the aneurysm, which eventually ruptures [3].

Lipid metabolism encompasses the biosynthesis and degradation of lipids, including fatty acids, triglycerides, and cholesterol. Specialized lipoproteins facilitate the transportation of lipids from the intestine to the liver (the primary site of lipid conversion) and between the liver and peripheral tissues. Lipid metabolism disorder (LMD) is a common condition characterized by abnormal blood levels of lipids or lipoproteins. LMD is a well-established risk factor for systemic atherosclerosis and cardiovascular diseases [4,5]. However, the role of LMD in IA formation and rupture is less clear [6,7,8]. It has been increasingly investigated in recent years. The formation and rupture of IA is a complex multifactorial process involving genetic, environmental, lifestyle, and other factors [9,10,11]. Among them, LMD could affect the progression of IA through several mechanisms, such as inducing systemic inflammation and oxidative stress, altering the lipid composition and metabolism of the intracranial artery wall, weakening the structural strength and elasticity of the intracranial artery wall, and modulating the expression and activity of various genes, proteins, and related signaling pathways [12]. As depicted in Figure 1, intracranial aneurysms are influenced by LMD at different stages of their development and rupture.

In addition to its potential role in IA formation, LMD could also be related to the severity and prognosis of SAH after IA rupture. LMD affects cerebral blood flow, cerebrovascular reactivity, cerebral autoregulation, and collateral circulation in patients with SAH [13]. This change could influence the occurrence and outcome of cerebral ischemia, vasospasm, delayed cerebral ischemia (DCI), and early brain injury (EBI) [14,15,16]. Moreover, LMD could modulate the inflammatory response, oxidative stress, blood-brain barrier disruption, neuronal injury, and neurogenesis after SAH, which can affect the recovery and regeneration of brain tissue [17].

Therefore, exploring the relationship between LMD and IA and its potential role in biomarker search and treatment is of great significance for understanding the pathogenesis and development of IA, improving early diagnoses and intervention, and reducing the incidence and mortality of SAH. In this article, we aim to discuss recent advances in the correlation between LMD and IA and to propose future research directions. The main contents of this article include (1) reviewing the altered lipid environment associated with vascular diseases; (2) analyzing the relationship between LMD and IA formation; (3) analyzing the relationship between LMD and IA rupture; (4) analyzing the relationship between LMD and SAH after IA rupture; and (5) discussing the possibility of LMD as a biomarker and therapeutic target for IA.

## 2. Dysregulated Lipid Metabolism in Vascular Diseases

### 2.1. Fatty Acyls

Fatty acyls are chain-like carboxylic acids composed of carbon, hydrogen, and oxygen and are the basic units of lipid metabolism. Depending on the chain length and degree of unsaturation, fatty acyls can be classified into short-, medium-, long-, and extralong-chain types, as well as saturated, monounsaturated, and polyunsaturated types [18]. The effects of fatty acyls on vascular endothelial cells (VECs) and vascular diseases mainly depend on their structural characteristics and metabolic pathways. In general, saturated fatty acyls such as palmitic acid and stearic acid adversely affect VECs, while unsaturated fatty acyls such as oleic acid, linoleic acid, arachidonic acid, alpha-linolenic acid, eicosapentaenoic acid (EPA), and docosahexaenoic acid (DHA) have protective effects on VECs [19,20,21].

Saturated fatty acyls induce various abnormal phenomena in VECs, such as oxidative stress, inflammatory response, endoplasmic reticulum (ER) stress, apoptosis, and autophagy, thereby impairing VEC barrier function, increasing permeability, promoting leukocyte adhesion and migration, and leading to atherosclerosis formation [22,23,24]. These mechanisms include activating Toll-like receptor 2/4 (TLR2/4), nuclear factor-kappa B (NF-κB), c-Jun N-terminal kinase (JNK), and protein kinase C (PKC), as well as inducing NLR family pyrin domain containing 3 (NLRP3) inflammasome assembly and activation [25,26,27,28]. In addition, saturated fatty acyls can also alter VEC biological characteristics by affecting membrane fluidity, protein palmitoylation, mitochondrial function, and other aspects [29].

Unsaturated fatty acyls have the opposite effects. They protect VECs from oxidative stress, the inflammatory response, ER stress, apoptosis, and autophagy damage, thereby maintaining VEC barrier function, reducing permeability, inhibiting leukocyte adhesion and migration, and preventing atherosclerosis development. These mechanisms include activating peroxisome proliferator-activated receptor alpha/gamma (PPARα/γ), AMP-activated protein kinase (AMPK), sirtuin 1 (SIRT1), and other signaling pathways, as well as inhibiting NLRP3 inflammasome assembly and activation [30,31,32]. In addition, unsaturated fatty acyls could also improve VEC biological characteristics by affecting membrane fluidity, protein phosphorylation, mitochondrial function, and other aspects [33,34,35].

### 2.2. Triglycerides

Triglycerides are the predominant form of lipid storage in the body and are transported by lipoproteins in the plasma. Elevated triglyceride levels are associated with an increased risk of cardiovascular disease, especially in the presence of low high-density lipoprotein (HDL) cholesterol levels [36]. The mechanisms by which triglycerides and their associated lipoproteins contribute to atherosclerosis and thrombosis are not fully elucidated but could involve direct infiltration of the arterial wall, inflammation, oxidative stress, endothelial dysfunction, and impaired fibrinolysis [37]. Several genetic and environmental factors influence triglyceride levels, such as diet, alcohol intake, obesity, diabetes, insulin resistance, and medications [38]. Lifestyle modification and pharmacological therapy can lower triglyceride levels and potentially reduce cardiovascular risk [39]. However, the evidence for the benefit of triglyceride lowering on clinical outcomes is limited and inconsistent [40].

### 2.3. Glycerophospholipids and Sphingolipids

Glycerophospholipids, primarily residing in the cellular membrane, act as structural components, providing integrity and fluidity. They function as precursors for bioactive lipids such as diacylglycerols and phosphatidic acid, which participate in cellular signaling [41]. Importantly, certain glycerophospholipid species have been implicated in cerebrovascular disease, including stroke [42]. The generation of platelet-activating factor (a type of alkylacylglycerophosphocholine) in endothelial cells could contribute to the inflammatory response after ischemic stroke. This highlights the relevance of glycerophospholipid metabolism in stroke pathology [43]. On the other hand, sphingolipids, complex molecules derived from the aliphatic amino alcohol sphingosine, play key roles in cell signaling and recognition, endocytosis, and intracellular trafficking [5]. Ceramides, a group of sphingolipids, are considered bioactive lipids involved in the regulation of cellular processes such as inflammation, oxidative stress, and apoptosis. Elevated ceramide levels have been observed in brain tissue after stroke, which is postulated to contribute to neuronal cell death [44]. Moreover, the roles of sphingosine-1-phosphate (S1P) and its receptors in regulating blood-brain barrier integrity, neuroinflammation, and neuronal survival suggest the relevance of sphingolipid signaling in neurovascular diseases [45].

### 2.4. Apolipoprotein

LpA is a unique lipoprotein that consists of a low-density lipoprotein (LDL)-like particle and a glycoprotein called apolipoprotein A (ApoA). ApoA is synthesized in the liver and covalently linked to apolipoprotein B-100 (ApoB-100), the main protein component of LDL, with a disulfide bond. ApoA has a high structural similarity to plasminogen, the precursor of plasmin, which is an enzyme that dissolves blood clots. ApoA contains several repeating units called kringle domains, which are also found in plasminogen. The number and size of these kringle domains vary among individuals, resulting in different isoforms of ApoA and LpA.

Within the ApoA molecule, ten kringle IV subtypes could be identified. The heterogeneity in ApoA isoform sizes primarily arises from varying copy numbers of kringle IV type 2, ranging from a couple to over forty [46,47,48]. This intricate kringle IV type 2 repeat system creates a broad molecular weight diversity of LpA. The assembly of LpA remains an enigma, involving a series of intricate noncovalent interactions followed by the formation of a crucial disulfide bond [49,50]. Nevertheless, an inverse correlation is observed between plasma LpA levels and ApoA isoform size, primarily attributed to the constant production of ApoA in the liver, with smaller isoforms being generated more prolifically than larger ones [51]. Despite extensive research, the clearance pathways for LpA remain largely elusive, although LDLR-, VLDLR-, CD36-, and SR-B1-mediated hepatic clearance and proteolytic cleavage may play a role [52].

Through several mechanisms, LpA is suspected to be causatively linked with atherosclerotic cardiovascular disease (ASCVD). LDL-like particles bearing ApoA could instigate atherosclerosis, while the inherent properties of ApoA potentially exacerbate atherogenic risks. The binding sites of ApoA facilitate adherence to the damaged endothelium and subsequent particle entrapment in the subintimal space [53]. Additionally, oxidized phospholipids on ApoA could instigate plaque inflammation and atherosclerosis, with the propensity to promote endothelial dysfunction and drive inflammatory cascades [54].

An elevated LpA level has also been implicated in aortic valve calcification and is thus recognized as a risk factor for calcific aortic stenosis. These findings underscore the potential of LpA as a biomarker, since heightened plasma LpA levels are independently linked with increased ASCVD risk and calcific aortic stenosis. The largely genetically determined plasma LpA levels are influenced by the size of the LPA gene, a surrogate for the kringle IV type 2 copy number, and specific single-nucleotide polymorphisms [55]. Exploiting the presumed causative association between LpA and ASCVD, therapeutics such as pelacarsen, an antisense oligonucleotide targeting ApoA, are currently under development. As such, a deeper understanding of apolipoprotein biology will likely pave the way for future therapeutic strategies aimed at mitigating ASCVD risk [56].

## 3. Lipid Metabolism and IA Formation

Intracranial aneurysm formation is associated with genetic susceptibility. Emerging research has illuminated the potential connection between dysregulated lipid metabolism and the genetic susceptibility of IA. Recent Mendelian randomization studies provide compelling genetic evidence that underscores the intimate relationship between lipid levels and the development of IA. In a seminal study, Mendelian randomization was employed to explore the associations of modifiable lifestyle factors and cardiometabolic factors with the risk of intracranial IA. Notably, genetically predicted decreased physical activity, higher triglyceride levels, higher body mass index, and lower levels of low-density lipoprotein cholesterol appeared to be associated with a higher risk of IA and aneurysmal subarachnoid hemorrhage [57]. Subsequently, a study further revealed the effects of blood lipids and lipid-modifying drugs on IA via Mendelian randomization. They demonstrated that genetically determined HDL-C and LDL-C levels were associated with a reduced risk of IA and ruptured IA [58]. These findings accentuate the pivotal role of lipid metabolism in the pathogenesis of IA.

In particular, several key proteins and genetic variations, such as apolipoprotein (APO) and related mutations, have been implicated in the genetic susceptibility of aneurysms. Apolipoprotein E (APOE), a critical regulator of lipid metabolism, has been associated with the formation of IAs in some investigations. For instance, the study by Liu et al. revealed a significantly higher frequency of the *APOE* E2/E2 and *APOE* E2/E3 genotypes in IA patients than in healthy controls [59]. Such findings suggested that certain variations in the *APOE* gene could augment the risk of IA formation. Likewise, LpA has also been examined as a potential risk factor for IAs [60], and the results indicated that the average LpA levels were over twice as high in familial members with IAs compared to the control group, implying that elevated LpA levels may be implicated in IA formation [61,62,63].

Moreover, a study revealed alterations in the expression of several genes, such as *ABCA1*, *APOA1*, and *LDLR*, associated with lipid metabolism in patients with IA. Notably, an increase in the expression of the *LDLR* gene in IA patients underscores the potential role of lipid metabolism in the pathogenesis of IA. Furthermore, a specific genotype (A/G) and allele (A) of the *APOA1* gene were associated with an increased incidence of IA, whereas another genotype (G/G) and allele (G) indicated an opposite tendency. These findings substantiate the notion that genetic variations may significantly influence the development of IA [64]. Second, a study further emphasized the distinct impacts of lipid-metabolism-related genetic variations in intracranial and abdominal aortic aneurysms. The research demonstrated that genetic risk profiles associated with serum lipid levels and coronary artery disease (CAD) correlate with the risk of abdominal aortic aneurysms but not IAs. Conversely, the genetic risk profile associated with blood pressure correlates with the risk of IAs but not abdominal aortic aneurysms [65]. These observations suggest that a lipid-metabolism-associated gene could influence the formation and development of different types of aneurysms via diverse mechanisms.

## 4. Lipid Metabolism and IA Rupture

The role of lipid metabolism has come under scrutiny in the study of several factors contributing to IA rupture. Abnormalities in lipid metabolism could cause inflammation, oxidative stress, and regressive changes in the blood vessel walls, all seen as significant risk factors for IA rupture. First, arachidonic acid (AA) is a polyunsaturated fatty acid that can transform into a range of inflammatory agents. However, a study identified that AA metabolism in the unruptured IA wall remains comparatively stable, possibly due to the relatively diminished oxidative stress and inflammation [66]. Next, the build-up of lipids plays a significant role in IA rupture. The foamy transformation of smooth muscle cells (SMCs) signifies lipid accumulation, as they take in and store an excess amount of lipids. Additionally, the anomalous expression of lipid-transporting proteins such as lipoproteins and ABCA1 in the wall of IA could influence the accumulation and clearance of lipids [67]. Accumulation of lipids and oxidized lipids, together with a drop in the plasma levels of antioxidative lipid antibodies and antiatherogenic proteins, could be linked to the regressive changes and rupture of the IA wall [8,68,69]. Regarding lipid factors, research has discovered that the concentration of LpA in the IA sac is correlated with an increase in the IA wall enhancement and could be connected to the regressive changes and rupture of IA [70]. Additionally, a high-fat diet could stimulate the advancement of IA by increasing serum cholesterol levels [71]. Finally, studies found that sex hormones such as testosterone could enhance the risk of brain vessel damage and IA rupture by impacting lipid metabolism, for instance, by lowering the levels of HDL in plasma [72]. These research outcomes not only fortify our comprehension of the mechanisms of IA rupture but also offer new potential approaches for preventing and treating IA, such as lessening IA by improving lipid metabolism.

## 5. Lipid Metabolism and aSAH

Following aneurysm rupture, there is a significant alteration in cerebrospinal fluid lipid metabolism, including an overproduction of bioactive lipid molecules such as inflammatory mediators and oxidative stress factors. These elements could further instigate cellular injury and vascular responses, amplifying brain damage. For instance, alterations in the metabolism of arachidonic acid could enhance the production of inflammatory mediators such as prostaglandins and leukotrienes [73,74,75,76,77,78,79,80]. Furthermore, lipid peroxidation and cerebral vasospasm showed a close relationship. Lipid peroxidation could induce cell membrane injury, which in turn triggers the contraction of vascular SMCs, leading to cerebral vasospasm. Lipid peroxidation products could also exacerbate cerebral vasospasm by impairing endothelial cell function, affecting vasodilation [81,82,83,84]. Last, the genotype of *APOE* could influence lipid metabolism following aneurysm rupture and the resultant patient recovery and outcomes. While certain studies have identified an association between the *APOE* ε4 allele and poor prognosis post-SAH, others have not found such a correlation [85,86,87,88,89]. This suggests that the influence and recovery following cerebral aneurysm rupture linked to the *APOE* ε4 allele may be complex and potentially affected by multiple factors. In summary, changes in lipid metabolism post aneurysm rupture, the relationship between lipid peroxidation and cerebral vasospasm, and the role of *APOE* all bear significant importance for understanding the biological mechanisms and prognosis of SAH. However, these areas necessitate further research and exploration.

## 6. Lipid Metabolism as a Biomarker and Therapeutic Target for IA

### 6.1. Lipid Metabolites as Biomarkers for IA

Aneurysmal subarachnoid hemorrhage (aSAH) is one of the major diseases that causes death and disability. Therefore, it is necessary to find biomarkers that can evaluate the therapeutic effect and prognosis, stratify aSAH patients, and objectively monitor the response to treatment.

Emerging research illuminates the integral role of lipid metabolism as a biomarker in predicting the onset and evolution of IA and prognosticating aneurysmal (aSAH) outcomes. Initially, lipid metabolism was implicated in the growth and development of IA. Markers such as F2-isoprostanes and F4-neuroprostanes, indicators of oxidative stress, have been underscored as being instrumental in the progression of IA [90,91,92]. Concurrently, compelling evidence links total cholesterol (TC) and HDL levels as independent prognostic indicators of mortality and delayed cerebral ischemia in the aftermath of aSAH [93]. Finally, the concentration of cysteinyl-leukotrienes (cys-LT) in cerebrospinal fluid (CSF) following aSAH has been found to correlate significantly with cerebral vasospasm [94]. This underscores the potential role of cys-LT, another product of arachidonic acid metabolism, in the etiology of vasospasm. In conclusion, lipid metabolism serves as a pivotal biomarker in IA and aSAH. It offers valuable insights into disease pathophysiology, informs predictive modelling of disease progression, and could potentially enhance patient outcome prognostication.

### 6.2. Lipid Metabolism Disorder as a Therapeutic Target for IA

The treatment of IA mainly relies on surgical or endovascular approaches, such as clipping or coiling, but these methods have certain risks and limitations and cannot prevent the recurrence or de novo formation of aneurysms. Therefore, finding effective pharmacological interventions to prevent or delay the formation, growth, and rupture of aneurysms is an important topic in the field of IA. Lipid metabolism disorder as a target in the treatment of IA and SAH has received much attention from many studies.

The therapeutic potential of statins has been validated across several domains, including but not limited to their capacity to mitigate the onset and progression of aneurysms. Numerous animal-based investigations have elucidated their efficacy. For instance, simvastatin has an inhibitory effect on the inflammatory response within the aneurysm wall of a rat cerebral artery aneurysm model [95]. Concurrently elevating the prevalence of endothelial cells and SMCs, as well as downregulating the expression and activity of metalloproteinase-2 and metalloproteinase-9, forestalls damage to and remodeling of the aneurysm wall. Similarly, rosuvastatin can enhance endothelialization at the aneurysm neck in a rat embolization model [96,97], augment the number of endothelial progenitor cells (EPCs), and restrict inflammation of the vascular wall. These findings suggested that statins could confer protection to or facilitate the repair of the aneurysm wall via a multitude of mechanisms, forestalling its expansion and rupture. Moreover, in the context of clinical trials, a wealth of studies point towards the ability of statins to enhance the prognosis of SAH patients. For instance, a study demonstrated that oral pravastatin (40 mg/d) significantly reduced the occurrence of cerebral vasospasm and ischemic neurological deficits [98]. Similarly, in a study of 592 SAH patients treated with clipping or coiling surgery, atorvastatin (20 mg/d) significantly diminished the incidence of cerebral vasospasm and cerebral infarction and enhanced the Glasgow outcome score at the 6-month mark [99]. Furthermore, in a study of 1214 SAH patients drawn from 10 randomized controlled trials who received varying types and doses of statins, there was a significant reduction in the incidence of cerebral vasospasm, ischemic neurological deficits, and cerebral infarction. However, no significant impact on the mortality rate or neurological function prognosis was detected [100]. These findings suggested that statins could enhance vascular endothelial function, resist oxidative stress, and inhibit vascular inflammation and coagulation, thus mitigating cerebral vasospasm and ischemic injury post-SAH. Statins, as a class of effective drugs to improve lipid metabolism, have shown some effects in animal experiments and clinical trials, but further research is needed to explore their optimal dose timing and indications as well as their possible adverse reactions and complications. As presented in Table 1, the enrolled clinical trials and subjects varied in terms of their basic characteristics, such as the study design, sample size, intervention, and outcome measures.

Eicosapentaenoic acid (EPA), an ω-3 fatty acid, has been identified to demonstrate considerable potential in the management of IA and SAH following aneurysm rupture. Its principal effects appear to reside in the inhibition of cerebral vasospasm (CV) and inflammatory responses. A study demonstrated that EPA significantly reduces the incidence of cerebral vasospasm and cerebrovascular infarction induced with CV in SAH patients [111,112]. A further study elucidated that the combined application of EPA and DHA, another ω-3 fatty acid, could markedly attenuate vasospasm, reduce the occurrence of new-onset cerebrovascular infarction due to spasm, and improve the clinical prognosis at 90 days post-SAH [113]. A study specifically investigated the impact of EPA on the progression of IA. Their findings suggested that EPA could significantly inhibit the size of IA in rats and degenerative changes in the media of arterial walls. Furthermore, EPA also appears to suppress inflammatory responses within the lesion, including macrophage infiltration and the expression of MCP-1, a chemokine primarily responsible for attracting macrophage accumulation in lesion areas [114]. However, these findings are largely based on small-scale or animal studies, and further large-scale, randomized controlled trials are required to validate the effectiveness and safety of EPA in humans.

## 7. Conclusions

Several studies have proposed possible biological mechanisms by which lipid metabolism disorders affect the development and rupture of intracranial aneurysms by selecting typical biomarkers. However, these studies did not systematically explore the molecular alterations caused by lipid metabolism disorders in intracranial aneurysms. In recent years, omics techniques have become effective tools for the discovery of novel biomarkers and biological pathways. However, there are no studies that have implemented proteomics, lipidomics, or combined multiple omics techniques to comprehensively detect systemic molecular changes caused by lipid metabolism disorders in intracranial aneurysms and to explore the complex connections between these molecules. Future studies should focus more on this direction to elucidate the critical role of lipid metabolism disorders in the development and rupture of intracranial aneurysms.

## Figures and Tables

**Figure 1 biomolecules-13-01652-f001:**
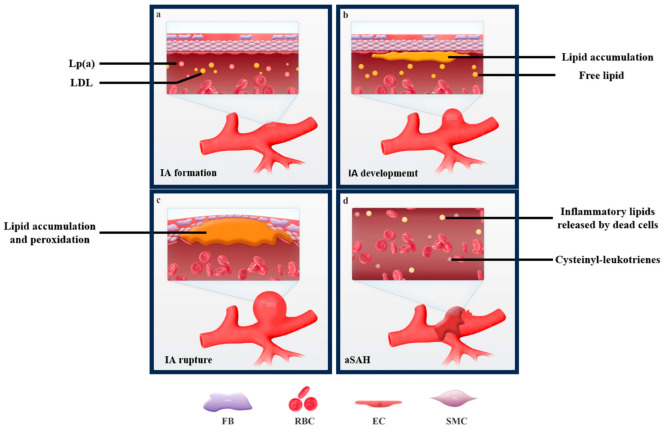
Intracranial aneurysms are influenced by LMD at different stages of their development and rupture (**a**–**d**).

**Table 1 biomolecules-13-01652-t001:** Basic characteristics of enrolled clinical trials and subjects.

References	Sample Size (E/C)	Diagnosis Standard	Experimental Group	Control Group	Treatment Duration	Outcome Measures
Tseng MY et al., 2005 [98]	40/40 aSAH	CT	Pravastatin 40 mg/d	Placebo	14 d	Postoperative complications (cerebral vasospasm, cerebral autoregulation, vasospasm-related DID)
Tseng MY et al., 2007 [101]	38/42 aSAH	CT/DSA	Pravastatin 40 mg/d	Placebo	14 d	Laboratory parameters, postoperative complications (DIND), clinical symptom scores (6-month mRS)
Kramer AH et al., 2008 [102]	71/79 aSAH	CT/CTA/DSA	Simvastatin 80 mg/d	Nonstatin	14 d	Postoperative complications (cerebral vasospasm, delayed infarction), clinical symptom scores (GOS of 1 to 3)
Vergouwen MD et al., 2009 [103]	16/16 aSAH	CT/CTA	Simvastatin 80 mg/d	Placebo	15 d	Laboratory parameters, postoperative complications, and clinical symptom scores (3- and 6-month GOS after SAH)
McGirt MJ et al., 2009 [104]	170/170 aSAH	CT/CTA/DSA	Simvastatin 80 mg/d	Nonstatin	14 d	Postoperative complications (cerebral vasospasm) and clinical symptom scores (perioperative death, length of hospital stay, discharged GOS)
Sanchez-Peña P et al., 2012 [105]	142/136 aSAH	CT/DSA	Atorvastatin 40 mg/d	Nonstatin	21 d	Clinical symptom scores (discharged and 1-yr GOS and modified Rankin Scale)
Naraoka M et al., 2018 [106]	54/54 aSAH	CT	Pitavastatin 4 mg/d	Placebo	14 d	Postoperative complications (DIND, cerebral vasospasm, vasospasm-related new cerebral infarctions)
Chen J et al., 2020 [99]	150/150 aSAH	CT/CTA/DSA	Atorvastatin 20 mg/d	Placebo	14 d	Postoperative complications (CVS, infarction, DIND) and clinical symptom scores (6-month GOS after SAH, 30-day mortality)
Li W et al., 2020 [107]	30/30 UIA	CTA/MRA/DSA/HR-MRI	Atorvastatin 20 mg/d	Placebo	12 M	AWE, aneurysm morphology, inflammatory factors (CRP, TNF-α, IL-1β, and IL-6)
Wang J et al., 2021 [108]	489/598 UIA	CTA/MRA/DSA	Atorvastatin 20 mg/d	Nonstatin	3 yrs	Aneurysm rupture (confirmed with CT or MRI)
Turhon M et al., 2022 [109]	20/20 UIA	CTA/MRA/DSA/HR-MRI	Atorvastatin 20 mg/d	Nonstatin	6 M	AWE, aneurysm morphology, inflammatory factors (CRP, TNF-α, IL-1β, and IL-6)
Kang H et al., 2022 [110]	30/30 UIA	VW-MRI	Atorvastatin 20 mg/d	Placebo	6 M	WEI, 3D-WEVR, aneurysm morphology, inflammatory factors

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
