# Peer review of "Intracranial Aneurysms and Lipid Metabolism Disorders: From Molecular Mechanisms to Clinical Implications"

_biomolecules, 2023, doi:10.3390/biom13111652_

Round 1

Reviewer 1 Report

Comments and Suggestions for Authors

Comments to Authors

I read d the manuscript of your review Intracranial aneurysms and lipid metabolism disorders: From molecular mechanisms to clinical implications

It is very interesting and sound correct with  reference to the issue. Considering the audience of this paper, It would be nice that:  1) Present a short paragraph and scheme of normal pathway; 2) then an scheme of points 2 and 3 to see how normal lipid metabolism is altered. 

Author Response

October 24, 2023

Dear Reviewer 1,

Thank you very much for your time involved in reviewing the manuscript and your very encouraging comments on the merits.

Comments:"It is very interesting and sound correct with  reference to the issue. "

We appreciate your clear and detailed feedback and hope that the explanation has fully addressed all of your concerns. In the remainder of this letter, we discuss each of your comments individually along with our corresponding responses.

To facilitate this discussion, we first retype your comments in italic font and then present our responses to the comments. 

Comment1: Present a short paragraph and scheme of normal pathway; then an scheme of points 2 and 3 to see how normal lipid metabolism is altered. 

Response1: We have supplemented the definition of normal lipid metabolism as per your request the modified part is marked in blue in line 46 to 50.

Reviewer 2 Report

Comments and Suggestions for Authors

Comments for author:

In this review, the authors summarized the role of abnormal blood levels of lipids or lipoproteins, which the author defined as lipid metabolism disorder (LMD), in IA formation, rupture and subarachnoid hemorrhage. The authors also discussed the possibility of using LMD as either a biomarker or the therapeutic target for IA.
This review is well organized, and the related literature is included comprehensively. The interpretation of the literature is mostly clear and sound. The references are appropriate.
Major points
1. In the introduction, the authors need to have more detailed descriptions on the process of IA formation. Although the process involves complicated processes, the authors at least need to describe the progressive changes occurring in IA wall and sac shown in Figure 1. This would provide the readers an overview of the different stages of IA. Alternatively, the authors can put the description into the figure legends of Figure 1.
2. In Figure 1, the square on top of the IA on the blood vessel has sides too thin to see clearly. The line that links Lp(a) to the red circle in the figure is also too thin. LDL, F2-isoprostane and cysteinyl-leukotriene are all shown as yellow spheres in different panels and are hard to differentiate. In panel b, the same color is use for F2-isoprostane and the lipid accumulation, which could potentially cause confusion that the majority of the accumulated lipid comes from F2-isoprostane. In panel d, does arachidonic acid, a quite hydrophobic fatty acid, exist in free form in the blood? In addition, arachidonic acid is not mentioned in Section 5 which discusses the relationship between LMD and aSAH.
3. Line 62, “reviewing the changed LMD associated with vascular diseases”. The term “changed LMD” is confusing. In section 2.1-2.4, the authors reviewed the role of different types of lipids and also lipoproteins in vascular diseases and how changes of their levels could affect vascular diseases. These may be better phrased as altered lipid environment.
4. In section 2.4, the reviewer recommends that the authors include a brief introduction of the structure of ApoA so that the readers can have a better sense how kringle IV subtypes are relevant to ApoA isoforms.
5. The usage of LpA and Lp(a) in the text is not consistent. Please correct them.
6. Section 3, line 176-188. The logic for this paragraph is not clear. In particular, the reviewer cannot see why the last sentence (Line 185-188) is an example of anything mentioned the text above it. Line 189, it is not clear what “this process” refers to.

Comments on the Quality of English Language

Author Response

October 24, 2023

Dear Reviewer 2,

Thank you very much for your time involved in reviewing the manuscript and your very encouraging comments on the merits.

We appreciate your clear and detailed feedback and hope that the explanation has fully addressed all of your concerns. In the remainder of this letter, we discuss each of your comments individually along with our corresponding responses.

To facilitate this discussion, we first retype your comments in italic font and then present our responses to the comments. 

Point 1: In the introduction, the authors need to have more detailed descriptions on the process of IA formation. Although the process involves complicated processes, the authors at least need to describe the progressive changes occurring in IA wall and sac shown in Figure 1. This would provide the readers 
an overview of the different stages of IA. Alternatively, the authors can put the description into the figure legends of Figure 1.

Response 1: We add a detailed description of the process of aneurysm formation, development until rupture as per your request, the modified part is marked in blue in line 33 to 46.

Point 2: In Figure 1, the square on top of the IA on the blood vessel has sides too thin to see clearly. The line that links Lp(a) to the red circle in the figure is also too thin. LDL, F2-isoprostane and cysteinylleukotriene are all shown as yellow spheres in different panels and are hard to differentiate. In panel b, the same color is use for F2-isoprostane and the lipid accumulation, which could potentially cause confusion that the majority of the accumulated lipid comes from F2-isoprostane. In panel d, does arachidonic acid, a quite hydrophobic fatty acid, exist in free form in the blood? In addition, arachidonic acid is not mentioned in Section 5 which discusses the relationship between LMD and aSAH.

Response 2: We have modified the picture and the labeling according to your request.

Point 3: Line 62, “reviewing the changed LMD associated with vascular diseases”. The term “changed LMD” is confusing. In section 2.1-2.4, the authors reviewed the role of different types of lipids and also lipoproteins in vascular diseases and how changes of their levels could affect vascular diseases. These may be better phrased as altered lipid environment. 

Response 3: We have corrected the mistake.

Point 4: In section 2.4, the reviewer recommends that the authors include a brief introduction of the structure of ApoA so that the readers can have a better sense how kringle IV subtypes are relevant to ApoA isoforms. 

Response 4: We have added introduction of ApoA as per your request, the modified part is marked in blue in line 148 to 155.

Point 5: The usage of LpA and Lp(a) in the text is not consistent. Please correct them. 

Response 5: We have corrected the mistakes.

Point 6: Section 3, line 176-188. The logic for this paragraph is not clear. In particular, the reviewer cannot see why the last sentence (Line 185-188) is an example of anything mentioned the text above it. Line 189, it is not clear what “this process” refers to. 

Response 6: In this paragraph, we want to discuss the relationship between lipid metabolism and gene susceptibility of  IA, and we have modified the statements in the article to make the discussion clear, the modified part is marked in blue in line 185 to 200.

Point 7: Line 220-222, “concentration of LpA in the IA sac is correlated with an increase in the IA wall”. What increase in IA wall? 

Response 7: Wall enhancement, we have corrected the mistake, the modified part is marked in blue in line 239 to 242.

Point 8: Line 283, please specify what is MMP.

Response 8: Metalloproteinase, we have corrected the mistake, the modified part is marked in blue in line 301 to 304.

Point 9: In Table 1, the reviewer recommends that the authors use lines to separate the rows. Otherwise, it is hard to tell which row the text in the Outcome column belongs to.

Response 9: Changes are made as per your request.

We would like to take this opportunity to thank you for all your time involved and this great opportunity for us to improve the manuscript. We hope you will find this revised version satisfactory. 

Sincerely,

The Authors
